# Protein–Mineral Composite Particles with Logarithmic Dependence of Anticancer Cytotoxicity on Concentration of Montmorillonite Nanoplates with Adsorbed Cytochrome *c*

**DOI:** 10.3390/pharmaceutics15020386

**Published:** 2023-01-23

**Authors:** Svetlana H. Hristova, Alexandar M. Zhivkov

**Affiliations:** 1Department of Medical Physics and Biophysics, Medical Faculty, Medical University–Sofia, Zdrave Str. 2, 1431 Sofia, Bulgaria; 2Institute of Physical Chemistry, Bulgarian Academy of Sciences (BAS), Acad. G. Bonchev Str., Bl. 11, 1113 Sofia, Bulgaria

**Keywords:** cytochrome *c*, montmorillonite, static light scattering, electric light scattering, microelectrophoresis, protein adsorption, in vitro anticancer cytotoxicity, cancer cell culture

## Abstract

Montmorillonite (MM) colloid nanoplates have high adsorption capacity due to their large size/thickness ratio, which allows them to be used as carriers for drug delivery. Upon adsorption of the mitochondrial protein cytochrome *c* (cytC) onto MM plates, the composite cytC–MM particles acquire anticancer properties because of the ability of cancer cells to phagocytize submicron particles (in contrast to the normal cells). In this way, exogenous cytC can be introduced into tumor cells, thereby triggering apoptosis—an irreversible cascade of biochemical reactions leading to cell death. In the present study, we investigated the physicochemical properties of cytC–MM particles as a function of the cytC concentration in the suspension, namely, the electrophoretic mobility, the mass increment of MM monoplates upon cytC adsorption, the ratio of the adsorbed to the free cytC in the bulk, the protein density on the MM’s surface, the number of cytC globules adsorbed on an MM monoplate, the concentration of cytC–MM composite particles in the suspension, and the dependence of cytotoxicity on the cytC–MM particle concentration. For this purpose, we used microelectrophoresis, static and electric light scattering, and a colon cancer cell culture to test the cytotoxic effects of the cytC–MM suspensions. The results show that the cytotoxicity depends linearly on the logarithm of the particle concentration in the cytC–MM suspension reaching 97%.

## 1. Introduction

### 1.1. Cytochrome c as a Proapoptotic Agent

Apoptosis is a programmed cascade of irreversible biochemical reactions leading to cell death that developed in the evolution of eukaryotic organisms as an innocuous (in distinction to highly toxic necrosis) mechanism for the suicide of their damaged cells. The process starts with the entry of the water-soluble mitochondrial protein cytochrome *c* (cytC, haemoproteid with molecular mass 12.4 kg/mol, 3 nm size, 104 amino acid residues, and one *c*-type haem-group) into the cytoplasm [1,2]; however, this does not occur in cancer cells because of their anaerobic utilization of glucose (instead of the mitochondrial oxygen metabolism that occurs in normal cells) [3,4,5]. Apoptosis in cancer cells can be initiated by the introduction of exogenous cytC via microinjection [6], but this laboratory technique is unadaptable to clinical therapy. The direct application of cytC solution is useless because the hydrophilic cytC globules cannot penetrate through the hydrophobic lipid bilayer of the cytoplasmic membrane. We apply another approach, which is based on the ability of cancer cells (an ability that is unlike that found in normal cells, except immune ones) to phagocytize extracellular particles with submicron size; this allows for the use of colloid particles as a protein carrier for the intracellular introduction of cytC previously adsorbed on the particles. This approach solves the main problem of anticancer chemotherapy, i.e., selectivity, due to the inability of health cells to phagocytose (they lose this capability in the process of differentiation in embryonic development).

### 1.2. Montmorillonite Nanoplates as a Drug-Delivery-Bearer

In order for the electrostatic adsorption of cytC to take place, its charge must be opposite to that of the colloid particles. As in other proteins, the charge of cytC is pH-dependent due to the ionization of the chargeable amino acid residues; the ratio of the carboxylic (COO^–^) and amino (NH_3_^+^) groups determines its isoelectric point, which emerges at pH 9 [7,8]. Since the cytC globules are positively charged at neutral pH (pH 6 is the most suitable because a buffer is not necessary), we have chosen montmorillonite (MM, a laminar alumosilicate clay mineral) whose charge is negative and pH-independent due to the isomorphic exchange of Al^3+^ in the central sublayer with extrinsic bivalent metal ions with lower valency (Zn^2+^ and Fe^2+^). The SiO-AlO-SiO crystal lattice has a flat form, 0.9 nm thickness, (sub)micro-metrically sized MM (nano)plates, and a large surface/mass ratio (about 250 m^2^/g) [9,10,11,12]; these properties facilitate the use of MM as a protein carrier.

### 1.3. Conditions for Proapoptotic Effectiveness

Apoptosis begins when the cytC concentration in the cytoplasm reaches a certain critical level, estimated as 16 μmol/L [13]. To achieve this concentration threshold, it is necessary to: (a) maximize the adsorption of cytC onto MM plates, and (b) increase the concentration of the composite cytC–MM particles in the cancer cell culture in order to achieve the phagocytosis of a larger number of cytC–MM particles. The number of cytC macromolecules adsorbed on an MM plate is confined by the maximal surface density of the protein globules and by the surface area of the plate. The protein density (the number of cytC macromolecules adsorbed on a given unit of MM area) is limited because of the 3 nm size of the cytC globules and the fact that they form only one monolayer upon saturated adsorption, as we have found previously [14]. The MM plate size must be restricted because of the inability of the cancer cells to uptake excessively small or large extracellular particles; the optimum is in the range 0.2–0.5 μm [15,16]. The two limits of the size are determined by the impossibility of the lipid membrane to envelop smaller particles (because of the high modulus of elasticity when bending occurs with a small degree of curvature) or larger ones (owing to a shortage of free membrane area). This means that it is advantageous to use MM plates with a size of about half a micrometer and to achieve saturated adsorption of cytC.

The second way to reach the threshold of cytC concentration in the cytoplasm (and by that to initiate the process of apoptosis) is to increase the concentration of cytC–MM composite particles in the extracellular medium in order to accelerate phagocytosis, whose velocity is determined by the opposite processes of endocytosis and exocytosis, as well as the velocity of protein destruction in the lysosomes under the action of protease enzymes. However, the particle concentration also has to be restricted because of the occurrence of aggregation when this process is increased (the aggregates cannot be taken up by phagocytosis owing to their excessively large size).

### 1.4. Unsolved Problems and Aims of the Investigation

Meeting the above two preconditions, we have achieved 95% cytotoxicity in a culture of colon cancer cells on the 96th h after the addition of a suspension of cytC–MM composite particles [17,18]. In terms of cytotoxicity, we achieved a result that is superior by 1/3, compared to the results reported by other authors (about 60–65%), by using various sophisticated artificial nanoparticles with adsorbed cytC [19,20,21,22,23,24,25,26,27,28]. However, unclarity remains regarding the number of protein globules adsorbed on the MM’s surface and how many remain free in the bulk of the suspension, considering that the non-adsorbed cytC has no anticancer effect (because of the impossibility to penetrate though the cytoplasmic lipid membrane). The ratio of adsorbed/free cytC globules and its dependence on the cytC concentration (at a fixed MM concentration) is important in order to optimize the preparation of a cytC–MM suspension when accounting for the price of the cytC, although we use the cheaper equine cytC whose 3D structure is almost identical to that of the human one [18 = SH-21].

Considering that phagocytosis is an active process that requires energy to cause membrane bending, it can be assumed that the cancer cells can uptake a limited number of particles for a certain time after the addition of the cytC–MM suspension to the cancer cell culture. Thus, it can be expected that reaching the intracellular cytC concentration threshold will initiate apoptosis; consequently, the degree of cytotoxicity would depend nonlinearly on the extracellular concentration of cytC–MM particles. The optimal concentration of cytC–MM suspension at which the cytotoxic effect is maximal is important with respect to its potential use in the clinical treatment of neoplasm.

The methods of increasing cytotoxicity considered above determine the purposes of the present investigation, which seeks to discover: (a) the ratio of adsorbed to free cytC and its dependence on the protein concentration; (b) the number of protein globules adsorbed on a unit surface and on one MM nanoplate; (c) the concentration of cytC–MM composite particles in the suspension; and (d) the dependence of cytotoxicity on the cytC concentration upon saturated protein adsorption. Therefore, we employ colloid physicochemical (microelectrophoresis, static, and electric light scattering) and biological (cancer cell culture) methods. The results disclose that the cytotoxic effect depends linearly in terms of a logarithmic scale on the concentration of the cytC–MM particles in the suspension.

### 1.5. Experimental Approach

As an indication for the adsorption of the positively charged cytC globules on the negatively charged MM monoplates, we employed electrophoretic mobility μ [m^2^s^–1^V^–1^] (particle migration velocity per unit field strength) using the free electrophoresis method in an aqueous medium. This method is based on the translational mobility of colloid particles in a direct electric field; to measure the value of μ, we used microelectrophoresis techniques in a dark optical field based on single-particles light scattering. The degree of mobility, μ, is determined by both the surface potential and the viscosity in the electric double layer (EDL). From the measured mobility μ, the electrokinetic potential ζ [mV] can be calculated, which is sensitive to the adsorption of counterions, thus reflecting the net charge density of the surface. However, the value of ζ cannot be calculated correctly because (in contrast to the small ions) the hydrodynamic flow is perturbed by the adsorbed cytC globules, whose 3 nm size is commensurable with the thickness δ [nm] of the EDL; this does not allow for the quantitative determination of the protein density.

Therefore, in order to quantify the number of cytC globules adsorbed onto an MM plate, and, accordingly, to achieve the aims of the investigation, we employed the static light-scattering method because it allows for the measurement of the mass increment of an MM monoplate upon the adsorption of cytC; this approach is applicable due to the extremely thin MM nanoplate, whose 1 nm thickness is three times smaller than the size of the cytC globules. Our approach is advantageous because it allows for the direct determination of the adsorbed protein amount, in contrast to the techniques used by other authors, wherein the concentration of the non-adsorbed protein in the supernatant is determined after the sedimentation of the colloid particles. Furthermore, this technique’s combination with the electric light-scattering method to measure the surface area of the MM plates allows us to calculate the protein density on the surface and the number of cytC globules adsorbed on one MM monoplate.

## 2. Materials and Methods

### 2.1. Materials

An aqueous solution of ferric equine heart cytochrome *c* (cytC) (Calbiochem, Cat. 250,600) was prepared by dissolving dry protein powder in triple-distilled water.

Na-montmorillonite K10 activated by acid treatment (Sigma-Aldrich, St. Louis, MO, USA) with surface area of 220–270 m^2^/g was suspended in triple-distilled water, and—after a 5 min treatment at 20 kHz with ultrasonic disintegrator (Techpan, Poland)—a low-polydispersity fraction was obtained by fourfold centrifugation [14]. The concentration of the obtained MM suspension was determined by weighing the dried residue, which had been dried in a microwave oven.

The cytC–MM suspension was prepared at pH 6 (at which the cytC globules are positively charged [7,8], which is ensured to condition their electrostatic adsorption on the negatively charged MM nanoplates) under continuous stirring for 30 min at room temperature by slow addition of the protein solution to the MM suspension to a final concentration of 10 mg/L cytC in 3 mg/L MM, at which ratio protein adsorption was saturated [14]). The pH of the suspension was slightly adjusted to pH 6.0 by the addition of a very small volume of concentrated KOH.

### 2.2. Static and Electric Light Scattering

The coefficient *R*_θ_ [dm^–1^] of light scattering at angle θ in a suspension of chaotically orientated colloid particles (MM or cytC–MM) with molecular mass *M* [g/mol] and concentration *N*_0_ [dm^–3^ = L^–1^] (number of particles in one liter), or weight concentration *c* [g/L] = *MN*_0_/*N*_A_ (where *N*_A_ is Avogadro number), is determined by the intensities [J·dm^–2^s^–1^ = W/dm^2^] of the scattered *I*_0_ and the exiting *I*_ex_ beams, and by the scattering volume Ω [dm^3^] disposed on distance *r* [dm] from the photoreceiver:*R*_θ_ = (*I*_0_/*I*_ex_) (*r*^2^/Ω) = [*HM*^2^*N*_0_*P*(θ)] = *cHMP*(θ),(1)
where the optical constant *H* = (4π^2^*n*_0_^2^/λ_0_^4^*N*_A_)(d*n*/d*c*)^2^ [mol·dm^2^/g^2^] is defined by the wavelength λ_0_ [dm] in vacuum and the refractive index increment d*n*/d*c*, which is determined by the difference Δ*n* = *n*_1_–*n*_0_ of the refractive indexes *n*_1_ of the particles and *n*_0_ of the liquid medium [29]. The intraparticle interference function (form-factor) *P*(θ) is defined by the form, the relative size *B*/λ of the particles with diameter *B* [nm], and the relative refractive index *n* = *n*_1_/*n*_0_ at wavelength λ = λ_0_/*n*_0_ [nm] in the medium [30].

The diameter, *B,* of the MM nanoplates was measured by the electric light-scattering method, which is based on the electrooptical effect (EOE) arising when colloidal particles are oriented in liquid medium, specifically, increment Δ*I* = *I*_E_ − *I*_0_ (in an electric field with strength *E* and in its absence) of the intensity, *I,* of light scattered by the suspension [31]. In the transient mode, after the electric field is switched off, the EOE decreases with the time *t* because of the particles’ disorientation:Δ*I*_t_/Δ*I*_s_ = exp(−*t*/τ),(2)
where Δ*I*_s_ and Δ*I*_t_ are EOE values at stationary orientation and at moment *t* after the field is switched off (*t* = 0), respectively; the relaxation time τ is defined as the time for *e*-fold decrease in Δ*I*_t_ starting from the steady-state value Δ*I*_s_ down to Δ*I*_t_/Δ*I*_s_ = exp(−1) ≈ 37%. The value of τ = 1/6*D*_r_ is determined by the rotational diffusion coefficient *D*_r_ = *kT*/*f*_r_ (defined by the rotational friction coefficient *f*_r_), which is conditioned by the particle size *B* and form (*f*_r_~*B*^3^) and the viscosity η of the medium at temperature *T*. In aqueous suspension (η = 1 mPa·s at 20 °C) of disk-like particles with diameter *B* the relaxation time is:τ = 1/6*D*_r_ = (2η/9*kT*) *B*^3^ = 59 × *B*^3^.(3)
where the relaxation time τ [ms] and the diameter *B* [μm] of disk-like particles are given in milliseconds and micrometers, respectively.

Light-scattering intensity *I*_0_ and EOE Δ*I* (Equations (1) and (2)) were measured with a photomultiplier in relative units (mV) using a computerized, home-made apparatus at fixed scattering angle of θ = 90° and a vertical electric field with strength *E* = *U*/*d* [V/cm] (induced by horizontal electrodes) whose vector is perpendicular to the horizontal scattering plane (determined by the exiting and scattered beams); the distance *r* and the scattering volume Ω (Equation (1)) are also constructively fixed. Sinusoidal electric impulses with frequency of 5 kHz and voltage of *U* ≤ 160 V produced by a Wavetek-185 functional generator and a Krohn-Hite-7500 amplifier were applied to the electro-optical cell (10 mL volume, *d* = 0.26 cm distance between two parallel platinum electrodes with 1 cm^2^ area).

### 2.3. Microelectrophoreses

The electrophoretic mobility μ [m^2^s^–1^V^–1^] of a colloid particle in liquid medium with viscosity η and dielectric permittivity ε_0_ε is proportional to its electrokinetic potential ζ [mV] according to Smoluchowski–Hückel–Henry’s equation [32,33]:μ = (εε_0_/η) *f*(*a*/δ) ζ,(4)
where Henry’s function *f*(*a*/δ) = 0.81 at *a*/δ ≈ 7.5 was calculated for spherical dielectric particles with diameter of 2*a* = 0.46 μm in a medium with ionic strength 0.1 mmol/L = 100 mM; the latter results in the δ ≈ 30.5 nm thickness of the electric double layer, EDL.

The time *t* [s] for migration of 20 + 20 single cytC–MM composite particles over a fixed distance *l* [μm] in direct electric field with strength *E* = 16 V/cm was measured using Mark II apparatus (Rank Brothers, UK) with a dark-field optical microscope and a closed, vertical, flat electrophoretic cell while focusing in both stationary planes (where the electroosmotic flow was zero) and reversing the field direction for every particle. Then, the electrophoretic velocity *v* = *l*/*t* [μm/s] was used to determine the electrophoretic mobility μ = *v*/*E* [(μm/s)/(V/cm)].

### 2.4. Cytotoxicity

The cytotoxic effect of cytC–MM suspension on culture of a metastatic colorectal adenocarcinoma cell line isolated from a lymph node (ATCC^®^CCL-227) was determined by counting dead and living cells after 96 h treatment. The initial cell density was 1 × 10^4^ cells/well after 3-day preliminary cultivation at 37 °C in 96-well plate with L-15 culture media and the addition of solutions of 10% fetal bovine serum albumin and 1% antibiotic-antimycotic (Gibco, Billings, MT, USA). After being triply prewashed with 0.15 M NaCl, 0.1 mL suspension of cytC–MM composite particles in 0.15 M NaCl medium was added to each well, and the culture incubation continued for the next 96 h. Then, a vitality test was performed by determining the degree of light absorption of a dye: after an additional triple-washing procedure, 50 μL 1% solution of human serum albumin in PBS buffer and 50 μM Trypan Blue with 2% final concentration were added to each well; the dye solution was washed after 3 min treatment and three fields of the cultures were then photographed under an inverse microscope with objectives ×10 or ×5 (about *N* = 220 or *N* = 1280 cells in the field, respectively). The dead (blue-colored) cells were counted in the samples with low cytotoxicity, and the vital (colorless) cells were counted at high cytotoxicity (because a part of the dead cells are fully or partly destroyed; Figure 1 in Ref. [18]). Cytotoxicity, (*N*_d_/*N*_v_) × 100 percent, was calculated as the ratio of three times the number of dead (3 × *N*_d_) to vital (3 × *N*_v_) cells. A 100% level of viability (0% cytotoxicity) was defined as a number of vital cells equivalent to 3 × *N*_v_ = 660 in the control sample with 0.15 M of NaCl added.

### 2.5. Computer Techniques

The surface electrostatic potential of cytC globules was computed with a program designed for determining protein electrostatics, Propka [34,35], using the atomic coordinates of the crystallographic structure of equine heart cytochrome *c* (protein data bank ID: 1HRC), and was then visualized by Chimera [36]. The structure of MM nanoplate was generated by multiplication of the elementary crystallographic cells of montmorillonite (bentonite) with formula Si_8_^IV^(Al_4y_Mg_y_)^VI^O_20_(OH)_4_M^+^_y_ (y denotes the exchanged atoms at isomorphous substitutions; Mineral data base).

## 3. Results and Interpretation

The main results and their interpretation are described in Section 3.1, Section 3.2, Section 3.3, Section 3.4, Section 3.5, Section 3.6 and Section 3.7. The advanced interpretations are discussed in Section 4.1, Section 4.2, Section 4.3, Section 4.4, Section 4.5 and Section 4.6.

### 3.1. Adsorption of cytC on MM Plates

The dependence μ(*c*) of the electrophoretic mobility μ of the cytC–MM composite particles on the concentration *c* of cytC in the suspension shows a decrease in μ to zero, and then an increase with an opposite sign (recharging upon saturated adsorption) (Figure 1). The course of this curve indicates the adsorption of the positively charged (at pH 6) cytC globules on the negatively charged MM plates; the sign of μ is negative below the isoelectric point (which appears at 5:3 mg/mg cytC/MM ratio) and positive above it (across an equivalent level of electrostatic adsorption). However, it is not possible to accurately calculate the electrokinetic potential ζ from the measured value of μ (and, consequently, to quantify the number of adsorbed cytC globules) because, in the case of globular proteins, the migration velocity of the particles in a direct electric field is determined by both the electric surface charge and the degree of increased liquid friction (the hydrodynamic stream is disturbed owing to the presence of the 3 nanometric cytC globules on the smooth MM surface; Section 4.1).

### 3.2. Mass Increment

In order to overcome the above difficulty regarding the quantitative determination of the protein density (number of cytC globules per a unit MM surface) from the electrophoretic mobility value, it is necessary to use a method in which the cytC–MM particles do not undergo hydrodynamic friction, i.e., when they are not moving in an external electric field. For this purpose, we employed the static light-scattering method, which is based on the growth of the mass *M* of the MM plates upon the adsorption of cytC; the use of Equation (1) allowed us to define the mass increment Δ*M* of the cytC–MM composite particles by measuring the light-scattering intensity *I*_0_ of the MM and cytC–MM suspensions.

From Equation (1), regarding the relative mass increment, it follows that
Δ*M*/*M*_0_ ≡ (*M*_1_ − *M*_0_)/*M*_0_ = (*R*_1_/*R*_0_)^1/2^ − 1,(5)
where the lower indexes 0 and 1 at the mass *M* of a single MM monoplate and the light-scattering coefficient *R* of the suspension denote the absence or presence of cytC, respectively, i.e., bare MM monoplates or cytC–MM particles with concentrations *N*_0_ and *N*_1_ (number of particles per unit volume of the suspension). This equation allows for the determination of Δ*M*/*M*_0_ in the absence of particle aggregation (*N*_1_ = *N*_0_); then, the rest of the quantities in Equation (1) remain unaltered because (a) the weight concentration *c*_0_ of MM is fixed in the experiment; (b) the form-factor *P*(θ) remains the same (whereas the thickness increases to 7 nm upon the adsorption of cytC on MM — it remains much smaller than the wavelength in the medium λ = λ_0_/*n*_0_ ≈ 400 nm); and (c) the optical increment d*n*/d*c* (and the optical constant *H*) cannot be altered noticeably since the refractive index of the proteins is commensurable with the *n*_1_ of dielectrics such as MM plates.

The measured twofold increase in the scattering coefficient *R* from *R*_0_ (without protein) to *R*_1_ (upon the addition of cytC) (Figure 2, *Insert*) could be caused by the following processes: (a) the aggregation of bear MM monoplates in pairs (model 2MM); (b) the bilateral adsorption of cytC globules on the MM monoplates in the absence of aggregation (model 2cytMM; in this case, the number of cytC–MM composite particles remains equal to that of bear MM monoplates: *N*_1_ = *N*_0_); and (c) a combined adsorption–aggregation mechanism (models 3cyt2MM and 4cyt2MM; the number of MM monoplates in the aggregates is two or more). The hypothetical models of composite particles and aggregates which emerge in result of these processes are shown in Figure 3. The experimentally found *R* = 2 *R*_0_ was predicted by the 2MM and 2cytMM models. The realization of the 2MM aggregation model was not possible because the bear MM monoplates do not aggregate (the suspension is stable; *R*_0_ does not alter with the time) due to the electrostatic repulsion between neighboring plates with a similar (negative) electric charge. The indication for the feasibility of the 2cytMM adsorption model is the decrease in the electrophoretic mobility, μ, of the MM plates at low cytC concentrations, *C*_cyt_, and the change in the μ sign from negative to positive (recharging caused by overequivalent electrostatic adsorption) at high *C*_cyt_ (Figure 1), which is caused by the adsorption of the positively charged cytC globules on the negatively charged MM monoplates. The same indication for cytC adsorption (a decrease in electrophoretic mobility, μ) favors the combined adsorption–aggregation models 3cyt2MM and 4cyt2MM (they predict the aggregation of single cytC–MM particles caused by a reduction in interparticle electrostatic repulsion); however, these two models predict 3.5- and 4-fold increases in the light-scattering coefficient *R*, i.e., *R*_1_ = 3.5 *R*_0_ and *R*_1_ = 4 *R*_0_, but the experiment yields *R*_1_ = 2.0 *R*_0_ (Figure 2, *Insert*). Adsorption–aggregation models with a larger number of MM monoplates in their cytC–MM aggregates are even less probable because they predict even higher increases in the light-scattering coefficient, for example, *R*_1_ ≥ 7 *R*_0_ and *R*_1_ ≥ 8 *R*_0_ upon the pair coupling of the models 3cyt2MM and 4cyt2MM, respectively. Therefore, the results obtained through the static light-scattering method show that the only realistic model is the 2cytMM model (cytC adsorption on MM monoplates without the aggregation of cytC–MM particles).

The results in Figure 2 show that the light-scattering coefficient increases 2.0 times upon the saturated adsorption of cytC on MM, which corresponds to a relative increment of Δ*M*/*M*_0_ = (*M*_1_ − *M*_0_)/*M*_0_ = 41% of the initial mass *M*_0_ of the MM monoplates. The interruption of the curves of *R*(*C*_cyt_) and Δ*M*(*C*_cyt_) denotes the concentration range of the coagulation instability of the suspension, where mass *M*_1_ growth is caused by both protein adsorption and aggregation of cytC–MM particles (then, *N*_1_ < *N*_0_); the aggregation emerges because of the disappearance of electrostatic repulsion in the isoelectric point. In our experiments, aggregation is avoided both by the low concentration of MM monoplates (*c*_0_ = 3 mg/L), which conditions a sufficiently large interparticle distance, and by protein concentration, which is remote enough to lead to *C*_cyt_ = 5 mg/L, at which the recharging point occurs (Figure 1).

### 3.3. Adsorbed Protein Fraction

The relative mass increment Δ*M*/*M*_0_ of the MM monoplates allows for the determination of the fraction of the adsorbed cytC macromolecules at a given protein concentration *C*_cyt_, accounting for the facts that (a) the mass, *M*_0_, of the bare MM monoplates’ growth to mass *M*_1_ because of protein adsorption and (b) the number *N*_1_ of cytC–MM particles in a unit volume of the suspension remains equal to *N*_0_ of the MM monoplates due to the absence of aggregation (Section 4.3).

Then, the weight concentration *C*_ads_ [g/L] of the adsorbed component of the total concentration *C*_cyt_ of cytC in the suspension can be expressed by the relative mass increment Δ*M*/*M*_0_ of a single cytC–MM particle and the weight concentration *c*_0_ [g/L] of the bare MM monoplates (before the addition of cytC):*C*_ads_ = (Δ*M*/*M*_0_) *c*_0_.(6)

The fraction φ_ads_ of the adsorbed cytC is defined as follows:φ_ads_ = *C*_ads_/*C*_cyt_ = (Δ*M*/*M*_0_) *c*_0_/*C*_cyt_(7)

Figure 4 shows the adsorbed cytC fraction φ_ads_ [%] (the share of the adsorbed to all cytC macromolecules) as a function of the total concentration *C*_cyt_ in the suspension, which is calculated by Equation (7). The increasing φ_ads_(*C*_cyt_) function at small cytC concentration (0–3 mg/L) indicates a positive cooperative effect caused by the association of cytC globules at their 2D concentrating on the MM’s surface upon adsorption. In the absence of such an effect, the curve φ_ads_(*C*_cyt_) should have an initial horizontal plateau (due to the unperturbed degree of adsorption when there is a sufficient amount of free surface area) where the value of φ_ads_ is determined only by the adsorption constant, and then a downward course at a higher degree of surface occupation when the electrostatic repulsion between the positively charged protein globules hinders their adsorption on the negative MM surface.

The adsorbed fraction reaches a maximum of 40% at 3 mg/L cytC in a 3 mg/L MM suspension (Figure 4). At this concentration, the degree of protein adsorption is near its saturated value (the relative mass increment Δ*M*/*M*_0_ is close to its maximal value at *C*_cyt_ ≥ 10 mg/L, Figure 2). Therefore, this protein/mineral concentration ratio (1:1 mg/mg cytC/MM) is optimal for the preparation of cytC–MM composite particles; at this point, the MM plates can bear nearly the maximum number of cytC globules. However, in the in vitro experiments with cancer cell cultures, we used suspensions with a 10:3 mg/mg cytC/MM ratio (only 13% adsorbed cytC) due to their better coagulation stability (Section 4.4).

### 3.4. Protein Density on Monoplate Surface

The protein density *d*_cyt_ = *N*_cyt_/*S* (the number *N*_cyt_ of cytC globules adsorbed on the surface with area *S*) can be calculated from the measured relative mass increment Δ*M*/*M*_0_, molecular mass *M*_cyt_ = 12.4 kg/mol of the cytC macromolecules, Avogadro’s number *N*_A_ = 6.02 × 10^23^ mol^–1^, the specific surface area *s*_0_ = 250 m^2^/g of the MM monoplates, the fraction φ_ads_ = *C*_ads_/*C*_cyt_ of the adsorbed cytC globules, and the weight concentrations *c*_0_ and *C*_cyt_ of MM and cytC, respectively:*d*_cyt_ = (φ_ads_*C*_cyt_*N*_A_/*M*_cyt_)/(*s*_0_*c*_0_) = (*N*_A_/*s*_0_*M*_cyt_) (Δ*M*/*M*_0_)(8)

The number of adsorbed cytC globules per liter of suspension is (*C*_ads_/*M*_cyt_) *N*_A_ [L^–1^], where *C*_ads_ = φ_ads_ *C*_cyt_ [g/L] is the weight concentration of the adsorbed cytC in the suspension with respect to the total concentration *C*_cyt_. Considering that the MM concentration in our experiment is *c*_0_ = 3.0 mg/L, the total surface area of the suspended MM monoplates is *s*_0_*c*_0_ = 0.75 m^2^/L; consequently, the protein density *d*_cyt_ [m^–2^] is:*d*_cyt_ = (*N*_A_/*s*_0_*M*_cyt_) (Δ*M*/*M*_0_) = 1.94 × 10^17^ (Δ*M*/*M*_0_).(9)

Equation (9) yields *d*_cyt_ = 7.76 × 10^16^ m^–2^ at *C*_cyt_ = 3 mg/L (before the recharging point) and *d*_cyt_ = 8.15 × 10^16^ m^–2^ at *C*_cyt_ ≥ 10 mg/L (saturated adsorption: Δ*M*/*M*_0_ = 0.42, Figure 2); these values are equal to 7.8 and 8.1 cytC globules adsorbed on 100 nm^2^ of the MM’s surface (Figure 5).

### 3.5. cytC Globules per MM Lamella

The number *N*_cyt,1_ = 2*d*_cyt_*S*_1_ of cytC globules adsorbed on one MM monoplate is determined by its bilateral surface 2*S*_1_ and protein density *d*_cyt_. Assuming that the MM monoplates are shaped as circular disks with a diameter *B* and unilateral surface area *S*_1_ = (π/4)*B*^2^, the combination with Equation (8) yields the following equation at a relative mass increment (Δ*M*/*M*_0_) (caused by cytC’s adsorption on the MM monoplates in the absence of particle aggregation in the cytC–MM suspension):*N*_cyt,1_ = (π/2) *d*_cyt_ *B*^2^ = (π*N*_A_/2) (*B*^2^/*s*_0_*M*_cyt_) (Δ*M*/*M*_0_).(10)

The relaxation time τ_0_ = 4.7 ms (Figure 6), which is measured electrooptically from the transient process of EOE (Δ*I*_t_ decay after the electric field is switched off, Equation (2)) in the aqueous MM suspension at 20 °C, corresponds to disk-like particles with diameter *B* = 0.43 μm according to Equation (3). The τ_0_ was measured after a steady-state orientation in a sinusoidal electric field with a sufficiently high strength such that all (large and small) particles achieve a high degree of orientation; this guarantees the determination of a mean value of size *B*.

Equation (10) can be rewritten according to the obtained value—given above—of the protein density *d*_cyt_:*N*_cyt,1_ = 5.64 × 10^4^ (Δ*M*/*M*_0_),(11)
where the numeric constant is calculated at diameter *B* = 0.43 μm of the MM disks with a specific surface area of *s*_0_ = 250 m^2^/g and cytC molecular mass *M*_cyt_ = 12.4 kg/mol.

Equation (11) yields *N*_cyt,1_ = 2.37 × 10^4^, i.e., about 23 700 cytC globules adsorbed bilaterally on one MM monoplate (Figure 5, right ordinate) upon saturated adsorption when the relative mass increment is Δ*M*/*M*_0_ = 0.42 (determined via Equation (5) from the measured light-scattering coefficient *R*).

### 3.6. Concentration of cytC–MM Particles

The concentration of cytC–MM composite particles (the number *N*_1_ per 1 liter, equal to the *N*_0_ of bare MM monoplates in the absence of particle aggregation) can be calculated from the total surface *s*_0_*c*_0_ of the MM monoplates with a specific surface area of *s*_0_ at weight concentration *c*_0_, divided by the bilateral surface area 2*S*_1_ = (π/2)*B*^2^ of the MM disks with diameter *B*:*N*_1_ = *s*_0_*c*_0_/(2*S*_1_) = *s*_0_*c*_0_/(π*B*^2^/2).(12)

With regard to the particle concentration *N*_1_, Equation (12) yields *N*_1_ = 2.58 × 10^12^ [L^–1^] (2.58 × 10^9^ composite cytC–MM particles per 1 mL) at *s*_0_*c*_0_ = 0.75 m^2^ total area of the MM monoplates with a specific surface area of *s*_0_ = 250 m^2^/g at weight concentration of *c*_0_ = 3 mg/L (used in our experiment), and bilateral plate area 2*S*_1_ = 0.29 μm^2^ for the disks with diameter *B* = 0.43 μm (determined electrooptically by measuring the relaxation time τ = 4.7 ms). Upon adding 0.1 mL of cytC–MM suspension (Section 2.4), 2.58 × 10^8^ cytC–MM particles were introduced in one well of the cancer cell culture. Assuming that the cell density is 1 × 10^4^ cells/well, it can be estimated that the particle/cell ratio is 2.58 × 10^4^, i.e., about 25,800 cytC–MM particles per one cancer cell in the surrounding medium.

### 3.7. Particle–Concentration Dependence of the Cytotoxicity

To investigate the dependence of the cytotoxic effect on the concentration of the cytC–MM composite particles, the standard nutritive medium of colon cancer cell cultures was substituted with 0.1 mL suspensions with increased concentrations of both components (up to 20 times higher in comparison with 10 mg/L cytC and 3 mg/L MM), which were prepared at a constant 10:3 mg/mg cytC/MM ratio (which facilitates saturated protein adsorption, Figure 2). Besides cytC–MM, the cell cultures were treated with solutions of cytC and suspensions of bare MM monoplates, whose weigh concentrations corresponded to those of the cytC–MM suspensions, which were all immersed in the 0.15 M NaCl aqueous medium. The viability of the control culture treated with a pure solution of 0.15 M NaCl (an isotonic medium whose osmotic pressure is close to that of cytoplasm) was taken as 100%; the viability of the control was 96% compared to the culture with a standard nutritive medium. Cytotoxicity *N*_d_/*N*_v_ was calculated as a percentage according to the ratio of the number *N*_d_ of dead cells (blue-colored because of the permeability of the damaged cytoplasmic membrane towards the dye molecules) to the number *N*_v_ of vital cells (uncolored due to the intact cellular membrane). The cytotoxicity of the pure 0.15 NaCl solution was assumed as 0%.

The concentration dependences of the cytotoxicity *N*_d_/*N*_v_ on the particle concentrations *N* (the number of cytC globules *N*_cyt_, bare MM monoplates *N*_0_, and cytC–MM particles *N*_1_ in 1 liter) are presented (in linear coordinates) in Figure 7. The cytotoxicity of the cytC solution is close to zero; this supports the assumption that cytC macromolecules cannot penetrate through the cytoplasmic membrane. The dependence of the cytotoxicity of MM suspensions is also negligible; this may be due to the non-uptake of MM nanoplates because they are not absorbed on the cytoplasmic membrane owing to electrostatic repulsion (both the membrane and the MM plates are negatively charged).

The cytotoxicity of the cytC–MM composite particles reached 97% on the 96^th^ hour after the substitution of the standard culture medium with the cytC–MM suspensions with increasing particle concentrations *N*_1_. The form of the curve 3 (a rapid slope at low *N*_1_, which then decreases with *N*_1_) suggests that the dependence corresponds to a logarithmic function *y* = log(*x*). This supposition was verified by the presentation of the data in semilogarithmic coordinates (the *Insert* in Figure 7); the linearity of *N*_d_/*N*_v_ = log(*N*_1_) confirms this assumption: the particle concentration dependence of the cytotoxicity is logarithmic.

## 4. Discussion

### 4.1. Electrokinetic Potential

The isoelectric point of native cytC is pI 9.3 [7,8]; thus, its globules are positively charged in an aqueous solution at pH 6 (the actual pH determined by the dissociation of H_2_CO_3_ at normal pressure of CO_2_). Therefore, the adsorption of cytC on the negatively charged MM plates can be indicated by the decrease in the electrophoretic mobility μ of the cytC–MM composite particles corresponding to the cytC concentration (curve 1 in Figure 1). The number of cytC macromolecules adsorbed on one MM plate with a known surface area (determined electrooptically by Equation (3) from the measured relaxation time τ) could be calculated if the net charge of a single cytC macromolecule (computed at a given pH [7]) is taken into account and by employing the Poisson–Boltzmann equation (in Gouy–Chapman theory of the double electric layer) to calculate the density of the surface charge from the electrokinetic potential ζ whose value can be obtained by Equation (4) from the measured electrophoretic mobility μ. However, in the case of globular proteins adsorbed on a smooth surface (such as the 3 nm cytC globules on the crystal MM nanoplates), mobility μ is determined by both the electric charge of the surface and the hydrodynamic friction (in contrast to the case of small ions). The ratio of charge/friction decreases with the degree of protein density, and upon saturated adsorption, the charge of the MM’s surface is almost fully shielded (then, the charge of the cytC–MM particles is determined completely by the net charge of the cytC globules) [7,8]. This complication does not allow for the quantitative calculation of the ζ-potential from the level of mobility μ (measured by microelectrophoresis, or by other methods based on the translational mobility of the particles in direct electric field) nor, accordingly, the determination of the surface density of the adsorbed cytC globules and their quantities per one MM monoplate.

### 4.2. Light Scattering Coefficient

The use of Equations (1) and (5) to measure the intensity of the exiting *I*_ex_ beam and the intensities *I*_0_ and *I*_1_ of the scattered beam in absolute values [W/m^2^] is unnecessary because the mass increment Δ*M* upon cytC adsorption is proportional to the ratio *R*_1_/*R*_0_ of the light-scattering coefficients of the suspensions with and without added cytC, and since the scattering volume Ω and its distance *r* from the photodetector are constant (constructively fixed); so, it is sufficient to measure *I*_0_ and *I*_1_ in relative units (millivolts in our device).

The determination (by the static light-scattering method) of the number of adsorbed cytC globules is possible due to the extreme thinness of the MM nanoplates (in the case of spherical colloid particles or thicker plates, it is impossible to measure a mass increment Δ*M* commensurable with the mass *M* because of the excessively small surface/mass ratio): upon the adsorption of the 3 nm cytC globules, the mass of the 1 nm thick plates has increased enough so as to measure the increment Δ*R*_θ_ of the light-scattering coefficient *R*_θ_ ~ *M*^2^ and, consequently, calculate Δ*M*/*M*_0_ using Equation (5). However, this technique has a disadvantage in which the *M* can increase because of both the protein adsorption and aggregation of the cytC–MM particles.

### 4.3. Coagulation Stability

The results obtained by static light scattering (Figure 2) confirm the occurrence of protein adsorption (in accordance with the electrophoretic mobility; Figure 1) and allow for the determination of the relative mass increment Δ*M/M*_0_ upon the adsorption of cytC on the MM plates. However, the quantitative interpretation is only correct when the increase in *R*_1_ is occasioned by cytC adsorption but in the absence of the concomitant aggregation of cytC–MM particles (the second cause of the growth of mass *M* in Equation (1)).

Aggregation emerges when the cytC concentration (the ratio of cytC/MM) reaches a value at which the net charge of the cytC–MM particles is near zero (because of the adsorption of the positive cytC globules on the negative MM surface); then, the electrostatic repulsion between the neighboring particles disappears and they aggregate. In our previous research [14], it was found that the isoelectric point appears at a 5:3 mg/mg cytC/MM ratio; then, quick aggregation occurs, and the suspension becomes visually turbid. The suspension remains stable outside of the concentration range of 4–7 mg cytC in 3 mg/L MM; this allows for the use of Equation (5).

Aggregation can appear even at an overequivalent level of protein adsorption (where there is an excess of cytC, the positive net charge should prevent aggregation), because, in the process of recharging the cytC–MM composite, particles undergo a transient state with zero charge. To avoid this, we employed a low MM concentration; consequently, the aggregation of cytC–MM particles did not occur because cytC adsorption is faster than the collision of the next particles due to the sufficiently large interparticle distance and the very different translational coefficients of the nanometric protein globules and the micrometric colloid particles.

### 4.4. Protein/Mineral Ratio

According to the protein concentration dependences of the mass increment Δ*M*/*M* (Figure 2) and the share of adsorbed cytC (Figure 4), a 1:1 mg/mg cytC/MM concentration ratio is optimal (Section 3.3). At this ratio, 40% of the cytC in the suspension is adsorbed on the MM monoplates and their surface is almost maximally occupied by the cytC globules; thus, a single cytC–MM particle introduces the maximal number of cytC macromolecules upon its uptake into a cancer cell. This 1:1 concentration ratio is only valid for MM monoplates; in the case of untreated MM particles (which contain patches of monoplates), this ratio will be much smaller.

The concentration ratio of 5:3 mg/mg cytC/MM is unsuitable as it allows the cytC–MM particles to aggregate, although an aggregate of *N* cytC–MM particles bears *N* times more cytC globules. However, cancer cells cannot resorb such large particles because phagocytosis requires a sufficiently high area of the cytoplasmatic membrane to envelop the colloid particle.

At higher concentration ratios (≥10:3 mg/mg cytC/MM), aggregation does not occur due to electrostatic repulsion between the cytC–MM particles, which are already positively recharged (Figure 1). However, most of the cytC remains free in the suspension and cannot be introduced into the cancer cells because the hydrophilic protein globules cannot penetrate through the hydrophobic bilayer of the lipid membranes, and a single cytC globule cannot be phagocytized owing to its excessively small size (the membrane module of elasticity does not facilitate bending with a small radius). Nevertheless, we have chosen a 10:3 mg/mg cytC/MM concentration ratio because it raises the coagulation stability of the suspension compared to a 1:1 ratio both because of the higher net charge (Figure 1) and because of the steric repulsion between the surfaces of the cytC–MM plates, which are completely covered with hydrophilic protein globules (Figure 5, *Insert*).

At the ratio of 10:3 mg/mg cytC/MM, only 13% of the cytC globules are adsorbed, and 87% remain free in the suspension (Figure 4). The concentration of the free cytC in the cytC–MM suspension (prepared in distilled water) remains almost unaltered upon the addition of a concentrated solution of NaCl (0.15 M NaCl is required to equilibrate extra- and intracellular osmotic pressure) and the replacement of the medium of the cancer cell culture with the cytC–MM suspension (Section 2.4). The unabsorbed cytC macromolecules probably do not remain free in the culture medium; being positively charged at pH 7.4 [7,8], they are adsorbed on the negatively charged cytoplasmic membranes of the cancer cells, but this does not affect cell viability, as we determined early via the addition of the cytC solution (without MM) [18].

### 4.5. Surface Protein Density

Assuming that the cytC globule is spherical with a diameter of 3 nm, the protein density (*d*_cyt_ = 8.1 globules per 100 nm^2^, calculated by Equation (9), Section 3.4) corresponds to 57% occupation of the MM surface. This estimation is far too rough because the shape of cytC is not spherical (Figure 5; *Insert*); however, 2/3 is a realistic value that corroborates the following suppositions: (a) adsorption occurs randomly (because protein globules approach the surface successively in the time); (b) the globules remain chaotically arranged (they are stationary on the solid MM surface, in contrast to the lipid membranes in a liquid-crystal state); and (c) the degree of surface occupation is less than the maximum because of the electrostatic repulsion between the positively charged neighboring cytC globules (theoretically, the level of surface occupation can reach 79% with a maximally dense monolayer packing of the spheres on a 2D surface).

The inaccuracy in the calculation of the number *N*_cyt_ of cytC globules adsorbed on one MM monoplate with a bilateral surface 2*S* (Section 3.5, Equation (10)) is the sum of the errors that appear in the measurement of: (a) the mass increment Δ*M*/*M*_0_ upon the adsorption of cytC; (b) the unilateral surface *S*; and (c) the protein density *d*_cyt_ (Equation (8)).

To measure the surface *S,* we assumed that MM monoplates have the shape of a circular disk with diameter *B* and employed the electric light-scattering method which is very sensitive to the particle size due to the cubic dependence τ~*B*^3^ of the relaxation time τ on *B* (Equation (3). This equation is quite applicable to disk-like particles such as MM, whose half-micrometer quasi-diameter is two orders of magnitude larger than its 1 nm thickness; the degree of protein adsorption does not change significantly at this ratio due to the nanometric size of the cytC globules.

Figure 6 shows τ_0_ = 4.7 ms for the bare MM monoplates and τ_1_ = 6.8 ms for the cytC–MM particles upon saturated adsorption; we used τ_0_ to calculate *B,* assuming that the increment Δτ = τ_1_ − τ_0_ = 2.7 ms is caused by the hydrodynamic friction of the cytC globules adsorbed on the smooth MM surface but not by aggregation of cytC–MM particles. The indications for the absence of aggregates are the unaltered form of the relaxation curve Δ*I*_t_/Δ*I*_s_ = *f*(*t*) after protein adsorption (in the presence of aggregates, the curve assumes a long ‘tail’) and the constancy of τ_1_ for the long period following the preparation of cytC–MM suspension (upon aggregation, τ increases with time because of the growth in size).

The measurement of Δ*M*/*M*_0_ by the light-scattering coefficient *R*_θ_~*M*^2^ when there is a chaotic particle orientation is sufficiently accurate due to its square dependence on the particle mass *M* (Equation (1)). The inaccuracy in the determination of *N*_cyt_ by Equation (10) mainly stems from the protein density *d*_cyt_ calculated by Equation (9), where we use the value *s*_0_ = 250 m^2^/g for the specific surface area, which is the mean of the range 220–270 m^2^/g given by the producer of MM. We used the same value for the calculation of the concentration of *N*_1_ cytC–MM (number of particles in a unit volume), which is equal to that of bare MM monoplates (*N*_0_ in Equation (1)) due to the absence of aggregation (Section 4.3).

### 4.6. MM as a Protein Barer

We have chosen MM as cytC carrier because it is a symmetrical, laminated mineral, and this allows it to be split into monoplates unlike asymmetrical alumosilicates such as kaolinite. Like other clay minerals, in their natural state, MM nanoplates are packed in laminar packs (tactoids) that can be split via high-temperature treatment with acids due to the substitution of exchangeable bivalent counterions such as Ca^2+^ (which compensate the negative charge of the monoplates) with monovalent H^+^ (and then by K^+^ upon neutralization with KON); as a result, the tactoids are split to monoplates because the presence of monovalent counterions is not enough to compensate their negative charge, and so the electrostatic repulsion between them prevails over the van der Waals attraction. In the case of kaolinite, this cannot be performed because kaolinite’s monoplates have a transversal permanent dipole moment (due to their asymmetric structure), which more strongly attracts the monoplates in the tactoid, together with van-der-Waals forces.

### 4.7. Adsorbed Protein Detection

Our approach of measuring the mass increment via the static light-scattering method is advantageous because protein adsorption is determined directly on the particles. Other authors have used techniques for the indirect determination of the amount of adsorbed protein by measuring its concentration in the supernatant after the sedimentation of the colloid particles, or after dialysis; such an approach is imperfect because the effective surface area (accessible for the adsorption of globular proteins) decreases as particles aggregate. We avoid aggregation by implementing the electrophoretic control of the surface electric charge (such that it is away from the isoelectric point) of the particles, and by the detection of possible aggregates with the electric light-scattering method, which is very sensitive to particle size, *B*, due to the cubic dependence of the rotational diffusion coefficient *D*_r_ ~ 1/*B*^3^, in contrast to dynamic light scattering based on the linear dependence of the translational diffusion coefficient *D*_t_ ~ 1/*B*.

### 4.8. Cytotoxicity Definition

In our previous articles [17,18], we defined cytotoxicity as the ratio of the dead cancer cells in the cytC–MM suspension to the vital cells in the culture with a standard alimental medium (which contains all the components needed for the normal physiology of living cells of this type). In the present investigation, we defined cytotoxicity as the dead/vital cell ratio using the vital cell number in the control sample in which the standard medium was substituted with aqueous solution of 0.15 M NaCl (isotonic medium). The reason for this new definition is that 0.15 M NaCl is also a medium in the suspensions of cytC–MM composite particles; thus, the influence of the medium is identically considered in both the sample with the anticancer agent and in the control. Our correction increases the level of cytotoxicity from 95% to 97%; we believe that this new calculation is more correct because of the following considerations.

It was presumed that, in a 0.15 M NaCl medium, the cancer cells are less viable (owing to the absence of the necessary substances), and some of the dead cells could be mistakenly attributed to the action of the anticancer agent instead of the medium. This supposition was confirmed by the vital cell number in the NaCl control being 4% less than that in the sample with the standard culture medium. This low percentage discloses that the studied colon cancer cells are surprisingly viable, even 96 h after the replacement of the medium.

It seems more reasonable to prepare the cytC–MM composite particles in the standard culture medium to avoid the adverse effects of the NaCl medium in the anticancer suspension on cell viability. However, such a culture medium has high ionic strength (owing to the presence of NaCl and other small ions) and this could lead to the aggregation of the MM plates upon the adsorption of cytC (Section 4.3). To avoid this, we prepared the cytC–MM suspension using an aqueous medium and added NaCl (to 0.15 M concentration) immediately before the substitution of the standard medium in the cancer cell culture with the cytC–MM suspension.

### 4.9. Cytochrome-Carrier Composite Particles

A decade worthy of note different artificial particles has been used as carriers of the proapoptotic protein cytC [19,20,21,22,23,24,25,26,27,28]. Most of them were synthesized and have a complex structure, and their cytotoxicity was tested in vitro using a cancer cell culture or in vivo via body tumors in the following chronological sequence (year of publication, substance, cancer (clear cell line), and maximal cytotoxicity (if given)): (1) (2010, polymeric nanoparticles, Breast cancer (MCF-7), 65% [19]; (2) (2013, dendritic multi-domain nanoparticles, Lung cancer (A549), 60% [20]; (3) 2014, Fe-Au hybrid nanoparticles (iron oxide core and gold nano-shell), Liver cancer (Hep G2), 59% [21]; (4) (2014, mesoporous silica nanoparticles, Cervical cancer (HeLa), 55% [22]; (5) (2015, calcium carbonate, Breast cancer (MCF-7), ≈35% apoptotic cells [23]; (6) (2018, hybrid iron oxide-gold nanoparticles (covalently bond cutC combined with chemotherapeutics), Liver cancer (HepG2, Huh-7D, SK-hep-1),—% [24]; (7) (2019, ferritin nanocapsules, Acute promyelocytic leukemia (NB4),—% [25]; (8) (2019, ssDNA coated pH-responsive gold nanoparticles with reversible aggregation at pH 5.5, Melanoma (B16F10 cells), ≈10% apoptosis, ≈75% photothermal effect [26]; (9) (2020, polymer nanomicelles, Lung cancer (A549 cells), <50% apoptosis [27]; and (10) (2021, silanized large-pore mesoporous silica guarded with gold nanoparticles, body tumor (HeLa cells), ≤80% [28].

The level of cytotoxicity we achieved (97%) with simple cytochrome-carrier particles (MM nanoplates with a pH-independent surface charge) is much higher than the levels of the authors cited above (about 60–65%), except for the particles with a pH-dependent charge and those carrying anticancer chemotherapeutics (combined with cytC), although the latter was achieved with very complex composite particles; in particular, the cytotoxicity of cytochrome–mineral composite particles [22,23,28] is significantly lower than ours. We suppose that the main reason for this is the different capabilities of the particles with respect to the adsorption (in preparation of the suspensions) and desorption (in the cancer cells) of cytC macromolecules. As a rule, the authors aimed to prepare very sophisticated complex particles; however, in most cases, this approach has not increased anticancer effectiveness. In contrast, we aimed to prepare cytC–MM composite particles whose size, form, and electric charge allow for the achievement of maximal cytC adsorption, uptake by phagocytosis and easy desorption in the cancer cells, according to the requirements formulated in our previous article [18].

## 5. Conclusions

The proapoptotic mediator cytochrome *c* (cytC) can be introduced into cancer cells via the phagocytosis of protein–mineral composite particles prepared by the electrostatic adsorption of positively charged (at pH 6.0) cytC globules on the negatively charged nanoplates of the clay mineral montmorillonite (MM). Concentrations of 10 mg/L cytC and 3 mg/L MM are optimal for the preparation of cytC–MM composite particles as they facilitate saturated adsorption and the absence of aggregation. The size of the MM monoplates used was 0.43 μm, as calculated from the rotational diffusion coefficient measured by electric light scattering. The numerical concentration of the composite cytC–MM particles of this size is 2.6 × 10^9^ per mL at a weight concentration of 3 mg/L MM. The protein–concentration dependences of electrophoretic mobility and the light-scattering coefficient disclosed that the particle net charge changed from negative to positive (overequivalent electrostatic adsorption) and the mass of the MM monoplates increases 1.4 times upon saturated cytC adsorption. The fraction of the adsorbed cytC macromolecules reached a maximum of 40% at a 1:1 mg/mg cytC/MM concentration ratio, and then decreased to 13% at a 10:3 mg/mg ratio, which was determined via static light scattering. Upon saturated adsorption, the protein density is about 8 cytC globules per 100 nm^2^, and one MM monoplate with a size of 0.43 μm bears about 23,700 cytC globules adsorbed bilaterally. With an increasing particle concentration up to 20-fold (at a constant 10:3 mg/mg cytC/MM ratio), the in-vitro anticancer cytotoxicity of the cytC–MM suspensions increases with logarithmic dependence, although the solutions of cytC and the suspensions of bare MM plates have no noticeable cytotoxic effect on the same colon cancer cell cultures.

## Figures and Tables

**Figure 1 pharmaceutics-15-00386-f001:**
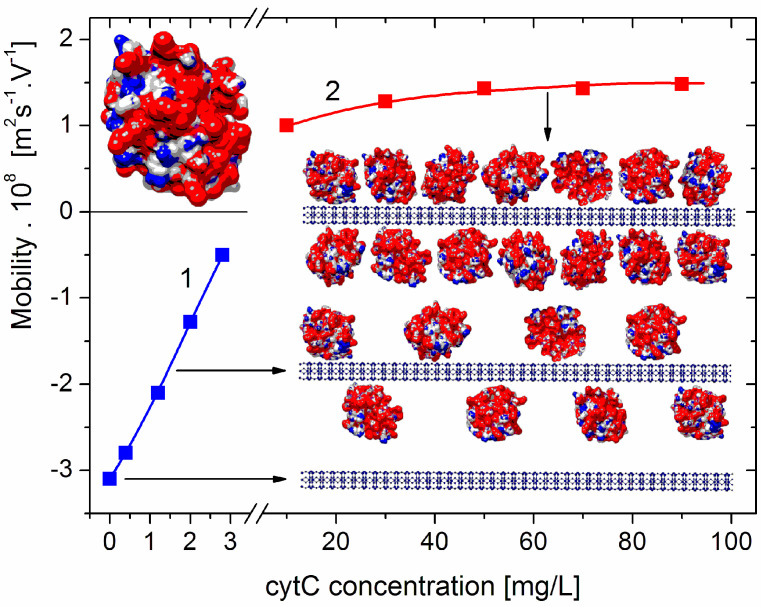
Electrophoretic mobility μ of composite cytC–MM particles at low (left part, curve 1, negative total charge) and high (right part, curve 2, positive total charge) protein concentrations. *Inserts*: Cytochrome *c* (cytC) globule (on the top left) and segments (on the right) of montmorillonite (MM) nanoplate (on the bottom: crystallographic structure, cross section, thickness of 0.9 nm, and negative electric charge) and two MM monoplates with adsorbed cytC globules (random orientation, positive net charge) at semi- and saturated levels of adsorption and actual 1:3 nm/nm ratio of MM thickness and cytC size. The cytC globules are colored according to the sign and value of their surface electrostatic potentials at pH 6 (positive—red, negative—blue, and neutral—white) with scale ranging from −6 *kT/e* to +6 *kT/e*, where *e*—the elementary charge (proton), *k*—Boltzmann constant, and *T*—absolute temperature; *kT/e* = 25.26 mV at 20 °C.

**Figure 2 pharmaceutics-15-00386-f002:**
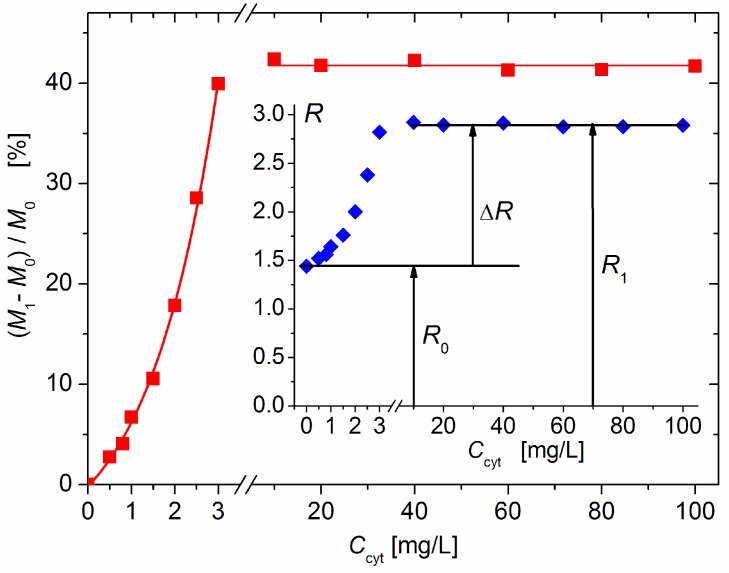
Relative increment (*M*_1_ − *M*_0_)/*M*_0_ in the mass *M*_1_ of composite cytC–MM particles upon adsorption of cytC on bare MM nanoplates with mass *M*_0_ vs. weight concentration *C*_cyt_ of cytC in 3 mg/L MM suspension. The dependence is interrupted in the protein range 3–10 mg/L because of particle aggregation. *Insert*: Protein concentration dependence of light-scattering coefficient *R* of 3 mg/L suspension of MM on the concentration, *C*_cyt_, of cytC, and the increment Δ*R* = *R*_1_ − *R*_0_ upon saturated adsorption.

**Figure 3 pharmaceutics-15-00386-f003:**
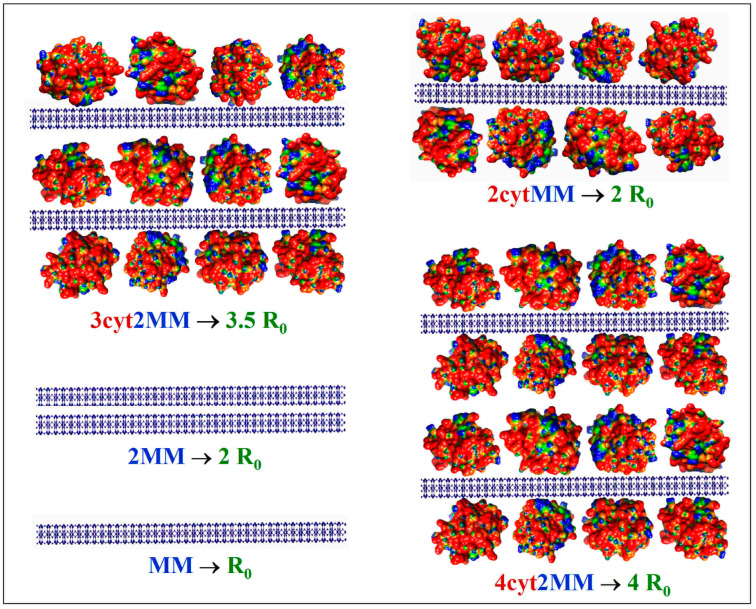
Hypothetical models of colloid particles in montmorillonite (MM) suspension with added cytochrome *c* (cytC) solution, which predict the corresponding increase in the light-scattering coefficient *R*_0_: single MM monoplate (MM); aggregate of two MM monoplates (2MM); single MM monoplate with two cytC monolayers adsorbed bilaterally (2cytMM); two MM monoplates with three cytC monolayers (3cyt2MM); two MM monoplates with four cytC monolayers (4cyt2MM). The cytC globules are colored in accordance with their surface electric potential (positive—red, negative—blue, and neutral—green) with respect to a scale of ±6 *kT/e*.

**Figure 4 pharmaceutics-15-00386-f004:**
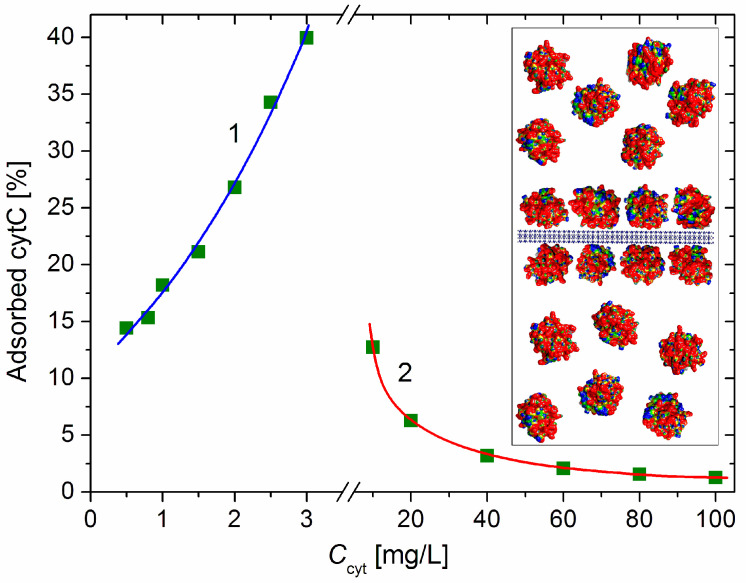
Fraction of the adsorbed cytC macromolecules vs. the concentration *C*_cyt_ of cytC in MM suspension with concentration *c*_0_ = 3 mg/L at low protein concentration (curve 1, negative net charge) and high cytC concentration (curve 2, positive net charge above the recharging point). *Insert*: Segment of MM nanoplate (cross section, horizontally oriented crystallographic structure, negative electric charge) with bilaterally adsorbed cytC globules (saturated protein adsorption at 1:1 mg/mg cytC/MM ratio) and free cytC globules in the surrounding medium at 4:6 adsorbed/free ratio (40% adsorbed cytC).

**Figure 5 pharmaceutics-15-00386-f005:**
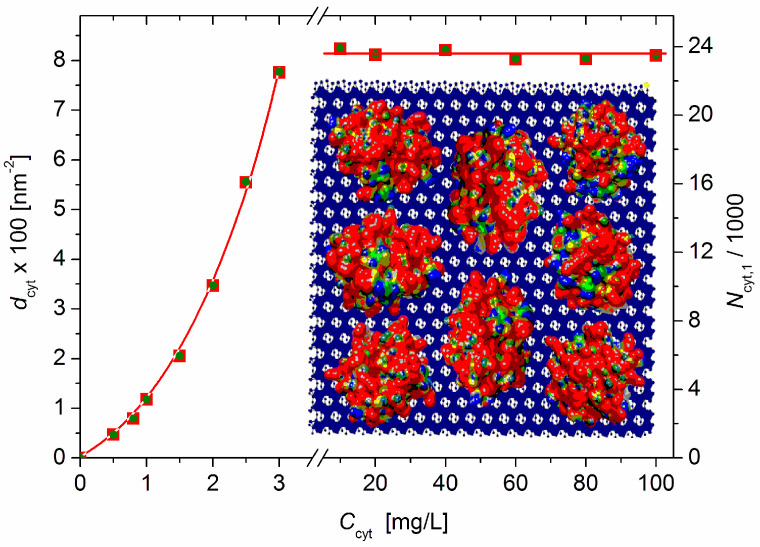
Protein density *d*_cyt_ (number of cytC globules adsorbed on 100 nm^2^ surface of MM monoplates) (left ordinate) and number of cytC globules adsorbed bilaterally on one MM monoplate with diameter *B* = 0.43 μm (right ordinate) vs. the concentration *C*_cyt_ of cytC in MM suspension with concentration *c*_0_ = 3 mg/L. *Insert*: Eight cytC globules with a size of 3 nm (with a random orientation and colored with scale ±6 *kT/e* according to their surface electric potential at pH 6: red—positive, blue negative, and green—neutral) adsorbed on 10 × 10 nm MM nanoplate (crystallographic structure. negatively charged) upon saturated adsorption.

**Figure 6 pharmaceutics-15-00386-f006:**
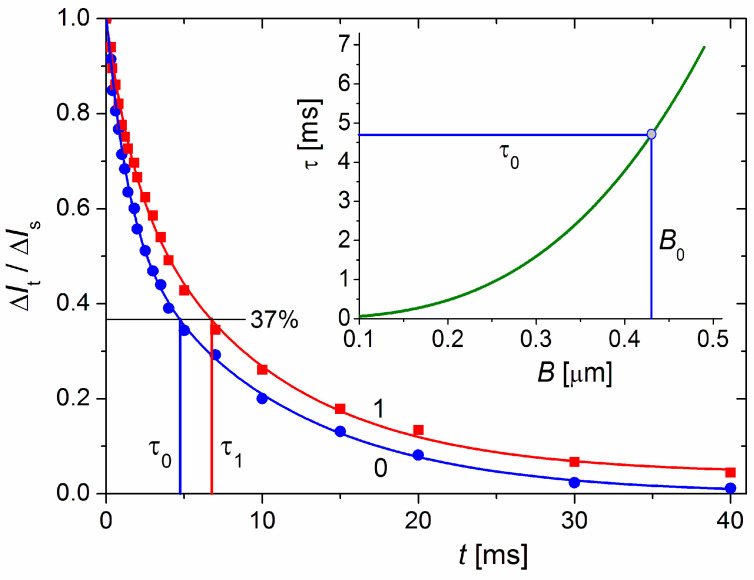
Time dependence of the dimensionless electrooptical effect Δ*I*_t_*/*Δ*I_s_* at moment *t* after switching off the electric field *E* in suspensions of bare MM monoplates (blue curve 0) and composite cytC–MM particles (red curve 1); the time *t* = 0 denotes the moment of switching off the field *E* after achieved steady-state Δ*I_s_* at high degree of orientation in sinusoidal electric field with frequency 5 kHz and strength *E* = 320 V/cm. The ratio Δ*I*_t_*/*Δ*I_s_* = 0.37 (denoted by black horizontal line) corresponds to values τ_0_ and τ_1_ [ms] on the abscise, respectively in the absence or presence of cytC in MM suspension with weight concentration 3 mg/L. *Insert*: Theoretical dependence of the relaxation time τ [ms] on diameter *B* [μm] of thin circular disks in medium with viscosity η = 1 mPa·s (water at 20 °C) according to Equation (3). The diameter *B*_0_ = 0.43 nm corresponds to τ_0_ = 4.7 ms measured for MM suspension.

**Figure 7 pharmaceutics-15-00386-f007:**
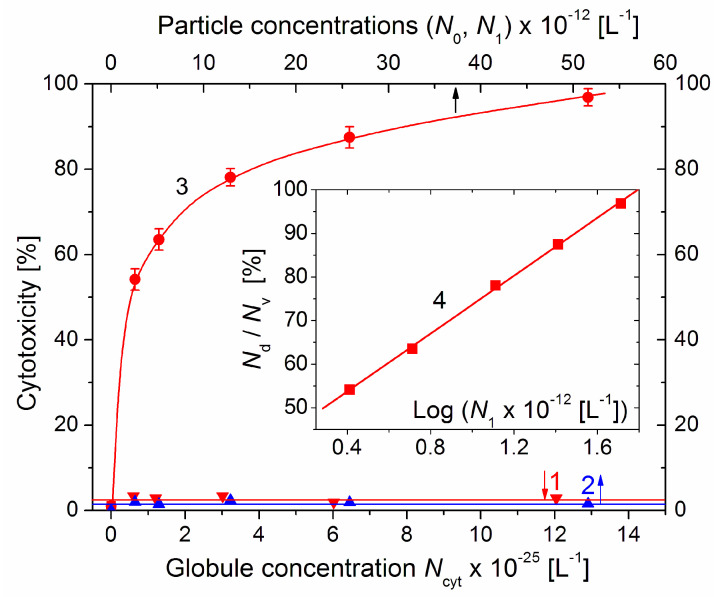
Cytotoxicity *N*_d_/*N*_v_ (dead/vital cells ratio) of cytC solution (line 1, lower abscissa), suspension of bare MM nanoplates (line 2, upper abscissa) and suspension of cytC–MM composite particles (curve 3, upper abscissa) on 96th hour after replacement of the standard nutritive medium in colon cancer cell culture with the corresponding solutions/suspensions vs. their particle concentrations *N* (number of protein globules *N*_cyt_, MM monoplates *N*_0_, and cytC–MM particles *N*_1_ in 1 dm^3^). The cytC–MM suspensions were prepared with concentrations of 10–200 mg/L of cytC and 3–60 mg/L of MM at constant 10:3 mg/mg cytC/MM ratio. *Insert*: Semilogarithmic dependence (line 4) of the cytotoxicity *N*_d_/*N*_v_ on the concentration *N*_1_ of cytC–MM composite particles (*N*_1_ = 2.58 × 10^12^ MM monoplates per liter at weight concentration of 3 mg/L MM).

## Data Availability

Not applicable.

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
