# Peer review of "Protein–Mineral Composite Particles with Logarithmic Dependence of Anticancer Cytotoxicity on Concentration of Montmorillonite Nanoplates with Adsorbed Cytochrome *c"

_pharmaceutics, 2023, doi:10.3390/pharmaceutics15020386_

Round 1
Reviewer 1 Report (Previous Reviewer 2)
I may recommend to accept this revised version for publication.
Reviewer 2 Report (Previous Reviewer 1)
The quality of the manuscript has been after revision improved (in some paragraphs substantially). The quality of the English is also much better than in the previous submission. In this situation, I can recommend to publish this article in the journal Pharmaceutics.
This manuscript is a resubmission of an earlier submission. The following is a list of the peer review reports and author responses from that submission.
Round 1
Reviewer 1 Report
The topic of the paper is in a principle appropriate for the publication in the journal Pharmaceutics. Although the idea behind this work is quite interesting (utilization of montmorillonite colloid nano-plates for adsorption and consequent delivery of cytochrome c to cancer cells), the quality of the results and discussion do not allow me to recommend this manuscript to be published in a highly impacted journal as Pharmaceutics.
Note: The title of the manuscript leads to a presumption that the article mainly deals with anticancer activity of the prepared complex. However, in the discussion there is no reference about the anticancer activity of the mineral-cytochrome c complex.
Moreover, although I am not a native English speaker, I have to say that numerous grammatical, stylistic, and formal errors are present in the manuscript. Several sentences are written in a style that it sometimes makes the understanding of the text very difficult.
Under above mentioned circumstances, I recommend submit this paper (after modification) to a journal more specialized on physico - chemical properties of drug carriers-protein complexes with a lower impact factor than Pharmaceutics.
Reviewer 2 Report
In this manuscript, Hristova and Zhivkov studied Protein-mineral composite nanoplates ( cytC-MM) possess anticancer effect using cultured tumor cell lines.
Comments:
In the Abstract, “At adsorption of the mitochondrial protein cytochrome c (cytC) on MM plates the composite cytC-MM particles acquire anticancer properties due to the capability of cancer cells to phagocytize submicron particles (in difference from the normal cells) …” Have you compared the effect on both normal and tumor cells? The data should be included in the Result section since this is an important experiment to show the selectivity of this cytC-MM suspension.
Effect on the induction of apoptosis should be measured as well.
You need to write a summary (conclusion) at end.